# Unveiling Arthropod Responses to Climate Change: A Functional Trait Analysis in Intensive Pastures

**DOI:** 10.3390/insects15090677

**Published:** 2024-09-07

**Authors:** Sophie Wallon, François Rigal, Catarina D. Melo, Rui B. Elias, Paulo A. V. Borges

**Affiliations:** 1CE3C—Centre for Ecology, Evolution and Environmental Changes, Azorean Biodiversity Group, CHANGE—Global Change and Sustainability Institute, School of Agricultural and Environmental Sciences, University of the Azores, Rua Capitão João d’Ávila, Pico da Urze, 9700-042 Angra do Heroísmo, Portugalrui.mp.elias@uac.pt (R.B.E.); paulo.av.borges@uac.pt (P.A.V.B.); 2Institut des Sciences Analytiques et de Physico Chimie Pour L’environnement et les Materiaux UMR 5254, Comité National de la Recherche Scientifque—University de Pau et des Pays de l’Adour—E2S UPPA, 64053 Pau, France; 3CFE—Centre for Functional Ecology, Universidade de Coimbra, Calçada Martim de Freitas, 3000-456 Coimbra, Portugal; 4IUCN SSC Atlantic Islands Invertebrate Specialist Group, 9700-042 Angra do Heroísmo, Portugal

**Keywords:** functional traits, arthropods, grassland, Azores, increased temperature, open top chambers, altitudinal gradient

## Abstract

**Simple Summary:**

Climate change is profoundly affecting ecosystem dynamics, with grassland arthropods serving as critical indicators. This in situ study examines the responses of arthropod communities to rising temperatures in intensively managed pastures on the volcanic island of Terceira (Azores, Portugal), employing a functional trait approach. Along an elevation gradient, Open Top Chambers (OTCs) were used to simulate increased temperatures, allowing the analysis of soil-dwelling arthropods over winter and summer seasons. Our results underscore the nuanced responses of arthropods to temperature variations across elevations and treatments within intensive pastures, revealing significant changes in functional community composition. This study emphasizes the importance of considering functional traits when assessing patterns of diversity in complex ecological communities involving multiple trophic levels.

**Abstract:**

This study investigates the impact of elevated temperatures on arthropod communities in intensively managed pastures on the volcanic island of Terceira, Azores (Portugal), using a functional trait approach. Open Top Chambers (OTCs) were employed to simulate increased temperatures, and the functional traits of ground dwelling arthropods were analyzed along a small elevation gradient (180–400 m) during winter and summer. Key findings include lower abundances of herbivores, coprophagous organisms, detritivores, and fungivores at high elevations in summer, with predators showing a peak at middle elevations. Larger-bodied arthropods were more prevalent at higher elevations during winter, while beetles exhibited distinct ecological traits, with larger species peaking at middle elevations. The OTCs significantly affected the arthropod communities, increasing the abundance of herbivores, predators, coprophagous organisms, and fungivores during winter by alleviating environmental stressors. Notably, iridescent beetles decreased with elevation and were more common inside OTCs at lower elevations, suggesting a thermoregulatory advantage. The study underscores the importance of considering functional traits in assessing the impacts of climate change on arthropod communities and highlights the complex, species-specific nature of their responses to environmental changes.

## 1. Introduction

Climate change has emerged as a defining challenge on a global scale, reshaping ecosystems and challenging the ability of different species to adapt [1,2]. In the context of these changes, arthropods as a taxonomic group are emerging as particularly sensitive indicators of environmental change [3,4]. These tiny but ecologically important organisms play a key role in maintaining ecosystem balance and are highly sensitive to changes in temperature, precipitation, and other environmental variables [5,6]. Thus, arthropods play a crucial role in ecosystem dynamics, reflecting the complex interplay between climate change and biological responses [7].

As temperature patterns shift and precipitation becomes more erratic [8,9], arthropods in different habitats are subject to disruptions in their abundance, distribution, and life cycles [10]. Understanding the impact of climate change on these arthropods is crucial not only for biodiversity conservation, but also for disentangling wider ecological consequences.

The global dimension of this issue enhances its importance. As indicators of environmental health, arthropods provide valuable insights into the wider effects of climate change [11]. Studying their responses to these changes can reveal patterns that transcend geographical boundaries, contributing to a more comprehensive understanding of the ecological consequences of climate change on a global scale [12,13].

Within this global panorama, however, our study focuses on a more specific investigation—the realm of pastureland arthropods in the volcanic archipelago of the Azores. Intensive managed pastures represent more than 56% of the land occupation in this region [14]. These islands host a unique diversity of arthropods, with most endemic species being currently restricted to mid-to-high-elevation native forests [12,13]. By narrowing our focus to a particular ecological niche (e.g., pastureland), we aim to unravel the nuances of climate change impacts on a specific group of arthropods associated with intensively managed pasture, providing insights that can be extrapolated to similar ecosystems worldwide.

The pastures in the Azores serve as habitats for various arthropod species from different origins (indigenous and non-indigenous) [15,16,17,18,19]. The interdependence between these arthropods and the grassland ecosystem is important for the good health of the pastures [20,21,22]. As climate change affects the Azores with temperature increases and changes in precipitation patterns (stronger events during the winter and longer periods of drought through the summer) [23,24], it poses unprecedented challenges for agriculture. In accordance with the findings presented in [14], the primary impacts linked to climatic factors in the Azores revolve around induced stress and diminished agricultural production during drought conditions. Additionally, there is notable evidence of reduced pasture and fodder quality, accompanied by the expansion of certain insect pests.

In the context of the Azores, temperatures are expected to increase by a value between 0.78 °C and 0.90 °C by 2039. Looking further ahead, to the end of the century, temperature projections indicate an increase of between 1.5 °C and 2.8 °C, corresponding to the two different scenarios, RCP4.5 and RCP8.5 [14]. 

The Azores, with their isolated and distinct ecological context [25,26], face both challenges and opportunities because of climate change. Shifts in temperature and precipitation pose a threat to the stability of arthropod communities [27], while, at the same time, opening avenues for understanding their resilience and adaptive strategies. 

To unravel the specific impacts of climate change on arthropods in the Azores, our study used Open Top Chambers (OTCs) as a controlled experimental tool along an altitudinal gradient. OTCs provide a unique opportunity to simulate temperature increases, allowing us to mimic expected climate scenarios [28]. First, this controlled environment allows for a focused study of how the arthropod communities within the pastures respond to elevated temperatures [29,30], a key facet of climate change. The use of OTCs aims to bridge the gap between observational studies and experimental manipulations [31,32], which should provide a more nuanced understanding of the direct effects of temperature change on arthropod functional traits in the Azores. Secondly, the inclusion of a small elevation gradient (180–400 m) in experiments on the effects of climate change on arthropod communities is crucial, as it provides natural variations in temperature and environmental conditions, allowing researchers to study species and community responses to these changes in a controlled setting [33,34]. The combination of OTCs and an elevation gradient is essential for understanding how climate change might alter biodiversity patterns and ecological interactions across treatments and elevations. 

Traditionally, species community monitoring has relied on the application of diversity indices based on taxonomic classifications [35]. These indices, such as the Shannon Diversity Index and species richness, provide valuable metrics for assessing the diversity and evenness of species within an ecosystem [36]. However, while taxonomic approaches provide a snapshot of community composition, they may miss subtle shifts in functional dynamics and ecological interactions [37]. Our previous study acknowledged the utility of these traditional metrics [30], but recognized the need for a more comprehensive approach to capture the nuanced responses of arthropod communities to climate change. By incorporating functional traits into our analysis, we aim to overcome the limitations of taxonomy-based assessments and provide a more holistic understanding of how arthropods are responding to changing environmental conditions in Azores pastures. This shift in methodology allows us to uncover functional nuances that may be obscured by traditional taxonomy-based alpha diversity measures (e.g., species richness, the Shannon Index), ultimately enriching our understanding of the ecological impacts of increased temperature on arthropod communities.

Functional traits, which include morphological and behavioral characteristics, provide a more dynamic lens through which to examine species’ responses to changing climate or disturbances [38,39,40]. The importance of this different approach lies in its ability to unravel how arthropod communities function within the ecological framework of Azores pastures under climate change. By focusing on functional traits, we can decipher not only which species are present, but also how they interact with their environment, providing a more nuanced understanding of the ecological consequences of temperature increase. 

We aim to determine how increasing temperatures (simulating climate-induced changes) along a short elevation gradient manifest themselves in a functional trait approach to arthropod communities within intensively managed pastureland. To this end, we seek to identify key functional traits that serve as indicators of adaptation or vulnerability in the context of temperature increase.

We posit that different guilds of arthropods within pastureland communities will display varying responses to temperature changes. Specifically, we anticipate that predator-and-prey relationships, as well as the presence of herbivores, will be reshaped under increased temperatures due to grass biomass availability. Arthropod guilds may respond differently to temperature shifts along the elevational gradient and between treatments due to their distinct ecological roles and specific traits. Predators, for instance, may experience changes in prey availability or altered hunting efficiencies (decreases in predators in the warmest sites: OTCs and lower altitudes), while herbivores may face shifts in plant suitability along the gradient, the treatment, and seasons [41].We hypothesize that as temperatures increase in pastureland ecosystems, there will be a specific pattern in the distribution of arthropod species’ body sizes, with smaller species in the warmest sites (OTCs and lower altitudes) [42].Beetles’ color and iridescence may influence their thermoregulation and capacity to support increased temperatures. Darker beetles should prevail in cooler sites, while iridescent beetles may be more abundant in warmer sites [43,44].

## 2. Materials and Methods

### 2.1. Study Area

The research was carried out on three experimental intensively managed pastures on the island of Terceira, part of the Azores archipelago in Portugal. The island of Terceira covers an area of 402 square kilometers and reaches a maximum altitude of 1023 m. Its geographical coordinates are between 38°37′ N and 38°48′ N latitude and 27°02′ W and 27°23′ W longitude. The sites, designated A, B and C, which are of low, middle, and high elevation, are located at different heights above sea level: site A at 186 m, site B at 301 m, and site C at 386 m. Their specific latitudes and longitudes are 38.703596° N, −27.353805° W for Site A, 38.701639° N, −27.325783° W for Site B, and 38.697770° N, −27.170075° W for Site C (Figure 1.). Each of these sites is classified as intensive pasture land. The primary vegetation consists of Italian ryegrass, *Lolium multiflorum* Lam. (Poaceae), in Sites A and B, and common velvet grass, *Holcus lanatus* L. (Poaceae), in Site C.

### 2.2. Experimental Design

A field experiment was established using Open Top Chambers (OTCs). OTCs are common tools used in climate change research to manipulate abiotic environmental conditions directly in the field to mimic an increase in temperature [28] (Aronson and McNulty 2009). The structure of OTCs includes panels that act as wind barriers, effectively reducing heat loss by convection. In addition, their open-top design allows rain to enter aronsand air to circulate freely, resulting in the creation of small air vortices or eddies [32] (Hollister et al. 2022).

Within each field, twenty plots measuring 1 × 1 m were arranged in a grid layout, while maintaining a distance of 1.5 m between them. Half of these plots, selected at random, served as a control group, while the remaining ten were enclosed by OTCs. To monitor temperature variations, data loggers (Easy Log: EL-USB-2) were installed both inside the OTCs and within the control plots. On average, temperatures within the OTC-enclosed plots were found to be 1.2 °C higher than those in the control plots. The OTCs were designed to encompass the 1 × 1 m plots, as well as an additional 25 cm border around each plot. This buffer area allowed for easy access to the plots for maintenance and monitoring without causing damage and facilitated the placement of pitfall traps at each outer corner of the plot, still within the boundaries of the OTCs. Additionally, the OTCs were elevated approximately 5 cm off the ground to permit the unimpeded movement of crawling arthropods throughout the entire sampling area.

Sampling took place in two different seasons, winter and summer 2020, and was carried out before any mowing of the grass. Cattle access to the sampling plots was not possible, since fences were implemented. Most importantly, the experimental design, which included both control plots and plots enclosed by OTCs, was maintained throughout the year. Consequently, there was no relocation of the OTCs and control plots between the two sampling periods.

### 2.3. Arthropod Sampling and Identification

In this research, we concentrated on the impact of intensive pasture management on arthropod communities, with a specific focus on crawling arthropods. This focus was necessitated by the fact that OTCs act as barriers to flying insects, potentially skewing results towards these ground-dwelling species. To facilitate sampling, pitfall traps were employed, specifically targeting epigean arthropods. Across each plot, four pitfall traps were positioned at every corner, resulting in a total of 80 traps per field. These traps, consisting of a 330 mL plastic cup with dimensions of approximately 12 cm in depth and 8 cm in diameter at the opening, were filled with a 20% solution of ethylene glycol from car cooling liquid and a few drops of soap to reduce water surface tension. To protect the traps from rain and prevent overflow, they were covered with a plastic dish secured by small iron sticks, allowing unobstructed access to the traps. The collected specimens were preserved in 96% ethanol.

For analysis purposes, the data from the four traps at each plot were combined into a single sample, yielding 10 replicates for each treatment type across the fields: ten for control and ten for OTCs. Arthropod collection was executed in the winter and summer of 2020, with the traps set for 14 days, except for one instance, in Field B, during summer, where the traps were set for 13 days—here, data for the 14th day were extrapolated based on the other days’ findings. However, species richness estimates were not extrapolated.

All collected arthropods were sorted and identified to the species level when feasible, covering target groups such as Arachnida (Araneae, Opiliones, Pseudoscorpiones), Diplopoda, Chilopoda, and most Insecta groups, including Formicidae and Lepidoptera, but excluding Diptera and Hymenoptera. Unidentified specimens were given a morphospecies code. The initial sorting and identification tasks were carried out by the lead author (SW) and assisting students (acknowledged as parataxonomists), followed by verification by an expert taxonomist (PAVB). Species nomenclature and colonization status followed the latest checklist of Azorean arthropods [45].

It is noteworthy that, in Terceira’s intensive pastures, exotic arthropods typically dominate, with native and/or endemic species’ presence diminished due to significant land disturbance [38,46]. Consequently, no distinction was made regarding the biogeographical origin of species in our analyses. Despite previous Azorean arthropod studies often including juvenile spiders (e.g., [46]), the prevalence of Erigoninae linyphiid spiders in this study posed challenges to adopting such an approach. All specimens are stored in the Dalberto Teixeira Pombo (DTP) Collection at the University of the Azores (Terceira Island), with data accessible in [18] through a specified URL.

### 2.4. Functional Traits

We gathered data on body size and dispersal ability for each arthropod species studied. We also compiled a collection of functional traits related to resource use. Finally, for beetles, specific traits were gathered regarding their color and presence or absence of iridescence (Table 1).

Information on traits was gathered through a thorough review of the literature, encompassing original species descriptions, initial records of species in the Azores, short communications, and ecological research. Additionally, information was obtained from experts who identified the specimens, or specialists (P.A.V. Borges, Pedro Cardoso, Sofia Terzopoulou) in the relevant taxonomic group, in cases where specific species information was unavailable.

Functional traits for each species were determined based on the characteristics of the adult stage. For morphospecies that could not be identified to species level, we assumed functional traits based on the closest taxonomic classification available, such as genus or family. Details of these trait assignments for each species can be found in Appendix A.

### 2.5. Data Analysis

Data from the four traps in each plot were grouped into a single value for analysis. During the winter, in four sampling events, one of the four traps was damaged due to flooding or earth filling. In these cases, we estimated the theoretical abundance of four traps by extrapolating from the data collected by the remaining three traps. Similarly, in cases in which traps were only operated for 13 days instead of the planned 14, we adjusted the sampling period by extrapolating arthropod abundances to cover the full 14 days. In situations in which either trap was damaged or the sampling period was reduced by one day, we assumed that species richness remained unchanged; only arthropod abundances were extrapolated.

Each trap was assigned a habitat, Field A, B or C, at low, middle, and high elevation, respectively, a treatment (control or OTCs), and a location (plot located on the edges or in the center of the field). All analysis were performed with R software (4.2.3) [47].

The analyses were carried out for winter and summer separately, considering all arthropods collected and only the most diverse taxon, namely the beetles (Insecta, Coleoptera).

For all arthropods, the traits analyzed were the following: the trophic guild (herbivorous, predators, fungivores, detritivores, and coprophagous organisms), how arthropods ingest food, e.g., feeding behaviors (piercing/sucking, chewing/cutting, and external digestion and sucking), some ecological traits, such as the daily activity (diurnal, nocturnal or active during twilight), the dispersal abilities (high or low dispersal abilities), and the standardized body size. 

Body size standardization was applied for each order present in our samples for analyses, considering all arthropods as follows:Standardized body size of Species *x* = (body size of Species × Order average body size)/Order Standard Deviation body size

For beetles, the traits analyzed were the following: the trophic level (herbivorous, predators, fungivores, and detritivores), the beetle’s coloration, the presence or absence of iridescence and some ecological traits, such as the daily activity (diurnal, nocturnal or active during twilight), high dispersal abilities, and the real body size. Feeding behaviors were not analyzed, as all beetles feed by chewing and cutting, and it was thus not a relevant functional trait to incorporate. Additionally, coprophagous trophic level and low dispersal abilities were not analyzed for beetles due to the lack of sufficient data to perform the analysis (only 3 species sharing those traits).

For each trait, we calculated the community weighted means (CWM) using the “functcomp” function from the FD package [48]. The CWMs express the mean attribute value between species occurring at a site, weighted by the relative abundance of each species [49], and were calculated as follows:$${CWM}_{j}=\sum_{i=1}^{S}p_{i}x_{ij}$$
where CWM*j* is the community-weighted mean value of trait attribute *j*, *p_i_* is the relative abundance of species *i* (*i* = 1, 2, …, S), and x*_ij_* is the value of trait attribute *j* for species *i*. Prior to the CWM computation, abundance data were square-root-transformed to reduce the influence of highly dominant species. Nominal traits were dummy-transformed to as many binary variables as there were trait attributes. All binary traits were then treated as continuous variables using the “bin.num” parameter in the function “functcomp”.

The resulting CWM values for both seasons were combined with habitat, location, and treatment information to form a comprehensive data frame. The relationships were visualized using boxplots created with the ggplot2 package, highlighting differences by field, location, and treatment conditions.

A series of analyses of variance (ANOVA) was conducted to investigate the effects of elevation, location, and treatment on each trait, including the interaction terms on each CWM variable and for each season. When the overall ANOVA model was statistically significant, post hoc pairwise comparisons were performed using the “lsmeans” package to determine differences between levels of significative factors and/or interaction. 

Preliminary tests indicated only four traits for all arthropods and four traits for beetles responding with low significances for the effect of the plot location (edges or center) with ANOVA, and, according to post hoc pairwise comparison, of the few traits responding to the effect of plot location, most were not significant. Thus, we restricted our ANOVA to the fundamental effects of our design, i.e., field, treatment, and interaction between the two.

Because several tests were performed for each season, we decided to correct the P-values using the False Discovery Rate method (FDR, [50]) to guard against inflation of Type-I errors.

## 3. Results

In total, over the three fields and two seasons (winter and summer), we collected 41,351 specimens. For the current analysis involving functional traits, only adults specimens were considered, giving a total of 35,735 arthropods from four classes, 15 orders, 60 families, and 171 morphospecies. From these, 34 taxa were identified only at the order, family, or genus level, resulting in 137 taxa with associated scientific species names (n = 32,821), which will be referred to as ‘species’ from here on (see full data in [18]). 

Our CWM analyses (Figure 2) show that the most important and strongest effect on all the functional traits studied is the factor field (e.g., elevation gradient) during both seasons for all arthropods and beetles. The second most important factor influencing functional traits is the treatment (control and OTC), i.e., elevated temperature. For all the arthropods, increased temperature had a greater effect during winter, especially on the ecological traits and guilds. Feeding behavior was not affected by the presence of OTCs. For the beetles, increased temperature had an effect on all the functional traits in both seasons. However, the effect was stronger in winter than in summer.

Finally, the interaction between field and treatment was the third main effect, and it showed that the effect of climate change varied according to the elevation. For all the arthropods, the interaction “field/treatment” showed an effect only in summer, mainly on feeding behavior and dispersal ability.

For the beetles, the interaction between field and treatment had a very small effect on diurnal activity, and for blackish beetles, this occurred in summer. In winter, color traits and diurnal and nocturnal activity were the most responsive traits. The results with the adjusted p-values were very similar, with only a few decreases with some significance. However, this did not change the main pattern of the results, confirming what was previously established.

With this context in mind, let us delve into the details. 

### 3.1. Winter and Summer Traits Analysis for All Arthropods

#### 3.1.1. Guilds

Considering all the arthropods during the summer, all the guilds (Figure 3) showed significantly lower abundances at high altitude than at middle and low altitude. There is an exception for predators, for which the highest abundance was found at mid-altitude for both seasons, showing a hump shape along the altitudinal gradient. During winter, the herbivores, coprophagous organisms, and fungivores also showed lower abundances at high altitude than at low altitude. However, the altitude gradient was less pronounced than in summer. Nevertheless, during winter, there was a treatment response for all trophic levels, with higher abundances in the OTCs, except for the detritivores, which showed no significant differences between altitude and treatment.

#### 3.1.2. Ecological Traits

With respect to ecological traits, during summer, low and high dispersers follow opposite trends, with a certain decrease along the altitudinal gradient for low dispersers and, inversely, an increase towards higher altitudes for high dispersers. During winter, high-dispersing arthropods also increase with altitude, but low-dispersing arthropods show no differences between fields. For both seasons, the OTCs showed a higher number of low-dispersal arthropods, while for the high-dispersal arthropods, the OTCs showed a lower number of individuals than the control plots. 

During the summer, the low-elevation field had the smallest body size community, while the medium-elevation field had the largest body size community. During winter, the arthropod body size increased with altitude, with larger arthropods at the highest altitude. Although there appears to have been a difference in body size between treatments in the high-altitude field during the summer, no significant differences were found (Figure 3 and Table A1).

In summer and winter, the diurnal arthropods decreased with altitude, while the nocturnal arthropods followed the opposite trend. In summer, the twilight arthropods decreased with altitude and in winter, they showed a hump shape along the altitude gradient with higher abundances at middle elevation. There were also fewer twilight and diurnal arthropods in the OTCs in winter.

#### 3.1.3. Feeding Behaviors

In terms of feeding behavior (Figure 3), chewing/cutting and piercing/sucking arthropods are less abundant at high altitude and show a slight decrease along the altitudinal gradient for both seasons. However, arthropods feeding by external digestion and sucking (e.g., spiders) are more abundant at the highest altitudes (Table A1 and Table A2).

### 3.2. Winter and Summer Beetle Trait Analysis

#### 3.2.1. Guilds

In terms of guilds, there was no clear pattern across all the guilds. Herbivores follow a hump shape along the altitudinal gradient in summer and a U shape in winter. Thus, the middle altitude field shows a large seasonal variation in herbivore beetles (Figure 4).

The predators appeared to be homogeneous at different altitudes, with no major changes along the altitudinal gradient. However, in winter, more predators were observed inside the OTCs.

Fungivores do not show major changes between different altitudes and seasons. 

The detritivores were more abundant during the winter, perhaps due to the higher humidity and availability of decaying matter. In addition, fewer detritivores were observed inside the OTCs during winter.

#### 3.2.2. Ecological Traits

The beetle body sizes showed a hump shape along the altitudinal gradient, with larger species at middle altitude in both seasons. Larger beetles were also found in the OTCs during summer (Figure 4). 

The number of diurnal beetles decreased along the altitudinal gradient in both seasons. In winter, fewer diurnal beetles were observed inside the OTCs.

The number of nocturnal beetles tended to increase with altitude, in addition to the fact that fewer nocturnal beetles were present inside the OTCs during winter.

The crepuscular beetles showed an opposite pattern between winter and summer. They decreased along the altitudinal gradient in summer and increased in winter.

Finally, the high-dispersal beetles showed a hump shape along the altitudinal gradient in summer, but a clear increase with altitude in winter.

#### 3.2.3. Color and Iridescence

During the summer, the beetles with orange and red shades (blackish and reddish brown) appeared to be less abundant at middle altitude and showed a U-shape along the altitudinal gradient. The highest abundance was found at low elevation (Figure 4).

Contrastingly, the black beetles were significantly more abundant at mid-altitude in both seasons, showing a hump shape along the altitudinal gradient. The lowest abundances of black beetles were observed at low altitudes.

Differences between treatments were observed for the black and reddish-brown beetles in both seasons. More black beetles were observed in the OTCs, while more reddish-brown beetles were observed in control plots.

Finally, the iridescent beetles showed a clear and significant gradient along the altitudes and during both winter and summer. More iridescent beetles were observed at low altitudes, while the lowest number of iridescent beetles was found at high altitudes. During winter, differences between treatments were observed, with more iridescent beetles inside the OTCs, especially at low elevation.

## 4. Discussion

The analysis of the functional traits across the three elevations and temperature treatments provides valuable insights into the effects of climate and environmental change on arthropod communities. The results of this study also highlight the complex and diverse responses of arthropods to short-term climate change, with distinct patterns emerging across different functional traits and seasonal conditions.

### 4.1. Influence of Elevational Gradient and Seasonality

#### 4.1.1. Guilds

The elevational gradient significantly influenced the arthropod and beetle communities, with noticeable variations between seasons. For all the arthropods, during the summer, guilds such as herbivores, coprophagous organisms, detritivores, and fungivores showed lower abundances at high elevations. This supports previous findings that arthropod diversity and abundance tend to decrease with elevation [51,52], potentially due to the optimal balance between temperature, precipitation, and other environmental conditions [53,54]. However, the patterns were not uniform across all the guilds, with the predators showing more homogeneous distributions along the gradient, albeit still with a peak at middle elevation. This confirms the finding in previous studies that different trends in the response of arthropod abundance along the elevational gradient depended on the level of taxonomic and functional resolution [53]. In a recent elevational study on Terceira Island native forest habitats, mixed results were also obtained, with the total species richness of all arthropods, Coleoptera, and Psocodea showing a monotonic decrease with elevation, but peaking at mid-high elevation for Araneae and endemic species [55].

During the winter, the guild analyzed with all the arthropods showed mostly differences between treatments, which will be discussed in the following section.

The beetles during the summer, herbivores, and predators also showed a peak at middle elevation, agreeing with [53,54]. The numbers of fungivores and detritivore beetles were very low during summer, probably due to the fact that their distribution is closely linked to the availability of organic matter and litter quality. However, litter quality depends mostly on water availability and temperature [56]. The lack of litter quality in summer explains the low abundances of those guilds. Detritivore arthropods play a crucial role in the decomposition of dead organic matter, including plant material and animal remains [57]. In our experiment, most of the detritivores were not strictly decomposers of the organic matter of either animal or plant origin, but of both. Thus, during the summer, the drop in their abundance was mainly due to the quasi-absence of litter, which represented a considerable part of the diet of our detritivores. During the winter, these organisms often exhibit increased abundance, potentially due to the higher humidity and availability of decaying matter [58]. Indeed, detritivore beetles were more abundant during the winter, which aligns with findings that cooler temperatures often slow down decomposition rates, leading to a prolonged availability of decaying matter [59,60]. The higher humidity during winter months may further support the activity and abundance of detritivores, as moisture plays a crucial role in maintaining their physiological functions [61].

#### 4.1.2. Ecological Traits 

Another particularly interesting finding was the varying responses of arthropods with different dispersal abilities along the elevation gradient. During the summer, low-dispersal arthropods tended to decrease in abundance at higher elevations, while high-dispersal arthropods showed the opposite trend, becoming more abundant at higher elevations. This trend suggests that the ability to move more widely provides an advantage for some arthropods and beetles, as they track shifting climate conditions, a pattern observed in various ecosystems for arthropods and other taxa [62,63]. These findings highlight the critical need to consider dispersal capabilities when examining the ecological impacts of climate change on arthropod communities. 

One more key result was the shift in the body size distribution of the arthropod community along the elevation gradient, with larger-bodied species becoming more prevalent at higher elevations, particularly during the winter season. This finding aligns with previous research demonstrating that larger-bodied arthropods often succeed in colder, higher-elevation environments, probably due to metabolic and thermoregulatory adaptations [42,64,65]. The prevalence of larger-bodied species at higher elevations during winter may reflect their ability to better withstand harsher conditions, while smaller species may struggle to maintain their energetic requirements

However, the beetles exhibited distinct ecological traits along the elevation gradient. Our data show that two of the largest beetle species were more abundant at the middle elevation than at the low or high elevations, e.g., the staphylinid, *Ocypus olens* (Müller, 1764), and the ground Beetle, *Calosoma olivieri* Dejean, 1831. Therefore, contrary to expectations, our results showed that larger beetle species tend to peak at middle elevations. This is likely to be due to the favorable conditions for growth and development found at these levels for grassland arthropods, where resource availability and environmental conditions are well-balanced. Arthropod abundance generally peaks at higher elevations on gradients with a very dry base and a sharp increase in precipitation with elevation [66]. However, the prolonged developmental time required by larger insects can be disadvantageous at higher elevations due to shorter active seasons [67]. Therefore, climatic conditions pose a significant obstacle to their optimal development and may limit their distribution along elevational gradients [68]. However, these patterns can vary, since some of the most iconic large Azorean endemic forest-adapted ground-beetles are restricted to high-elevation, hyper-humid forests [69].

The daily activity patterns of all the arthropods and beetles showed that the crepuscular and diurnal arthropods decreased with elevation, while the nocturnal arthropods increased, a trend that was consistent across both seasons. The altered patterns of activity across the diurnal, nocturnal and crepuscular arthropods along the elevational gradient and between seasons suggests that climate change could disrupt important temporal niche partitioning within arthropod communities. Furthermore, diel activity may influence arthropod responses to increased temperatures, with nocturnal species potentially being less affected by daylight heat compared to diurnal species. This is because nocturnal arthropods remain hidden from direct solar exposure and extreme temperatures [70]. However, our experiment showed opposite results to these findings, with more diurnal activity at lower elevations and more nocturnal activity at higher elevations. Species-specific responses and the complex interplay of environmental factors, including food availability and predation pressure, also influence diel activity [71], and could explain the results of our experiment. 

This shift in activity patterns may also be due to the thermal constraints imposed by elevation (harsher conditions and colder periods), which can influence the temporal niche partitioning within arthropod communities. As Lobo (2024) [72] states, “hotter is not better”, and some species of beetle with specific metabolic mechanisms are able to maintain their nocturnal activity under cold temperatures. Different metabolic mechanisms can make it difficult for diurnal beetles to remain active at cooler temperatures. The crepuscular beetles exhibited opposite patterns between winter and summer, decreasing in summer and increasing in winter, possibly due to shifts in temperature and resource availability that influenced their activity periods.

This highlights the complex ways in which climate change can impact the structure and dynamics of arthropod communities, with cascading effects on ecosystem function.

#### 4.1.3. Feeding Behaviors

The feeding behaviors were notably influenced by elevation, with the chewing/cutting and piercing/sucking arthropods decreasing at higher elevations, while those utilizing external digestion (e.g., spiders) increased. This may reflect adaptations to resource availability and climatic conditions at different elevations. Conversely, predators and scavengers that rely on external digestion, such as spiders (mostly linyphiids in the current study), may have found a higher and more consistent grass cover throughout the year at the high elevation in our experiment, providing a more favorable ecosystem, as shown by Borges and Brown (2001 & 2004) [73,74]. Our data also revealed a high number of harvestmen from the Leiobunidae family, which also rely on external digestion, at high elevation. Their high number could have been due to the higher moisture levels that they tend to prefer [75], which naturally occurs at this elevation. These adaptations highlight the complex interplay between environmental conditions and arthropod feeding strategies across altitudinal gradients.

#### 4.1.4. Beetle Color and Iridescence

The results indicate the significant presence of iridescent beetles along the altitudinal gradient, with more iridescent beetles observed at lower elevations compared to higher elevations. This pattern suggests that iridescence may be an adaptation to cope with the higher temperatures typically found at lower elevations and emphasizes the role of iridescence in thermoregulation. Indeed, iridescence in beetles can serve various ecological functions, including camouflage, signaling, and thermoregulation [76,77,78]. In fact, iridescent surfaces reflect a significant portion of incident light, which can help regulate body temperature by reducing heat absorption. Conversely, at higher elevations, where temperatures are generally cooler, the advantage of iridescence in thermoregulation might be less pronounced, resulting in fewer iridescent beetles [79]. It is noteworthy that all Azorean endemic beetles found in high-elevation, hyper-humid forests (at approximately 500 m to 1000 m above sea level), although not part of this study, are typically non-iridescent and mostly blackish. In our study, we did not collect endemic Azorean beetles, since their habitat is at higher elevation and the Azorean intensive pastures are mostly composed of introduced species and very few native ones. Therefore, it would be interesting in the future to study the coloration of native and endemic beetles in more detail to see whether their response follows the same trend as that of introduced beetles. The observed coloration trends along the elevational gradient in our experiment reveal significant ecological and adaptive patterns in beetle populations. During the summer, the beetles with orange and red nuances, such as blackish and reddish-brown beetles, exhibited a U-shaped distribution, with the highest abundances at low elevations. This pattern suggests that these colorations may be more effective for thermoregulation and camouflage in the warmer environments found at lower elevations [80,81,82]. Conversely, the black beetles displayed a hump-shaped distribution, being more abundant at middle elevations during both seasons. The adaptive significance of melanism in black beetles, particularly for absorbing heat in cooler environments, might explain their success in these mid-altitudinal zones [83].

### 4.2. Effects of Increased Temperature (OTC Experiment)

#### 4.2.1. Guilds

The increased temperature of the OTCs played a crucial role in the arthropod and beetle traits observed, especially during winter. The higher abundances of herbivores, predators, coprophagous organisms, and fungivores within the OTCs during winter suggest that increased temperatures may alleviate some environmental stressors, leading to increased activity and survival. Vasseur et al. [84] agree that an increase in temperature has these effects on ectotherms. However, they caution that it is important to consider the effect of the temperature mean and variance, as their model predicts that mid-latitude species will be most susceptible to large declines in performance under a future climate scenario when these two factors are considered. Furthermore, higher temperatures in OTCs during winter may reduce the energy costs associated with thermoregulation (for warming), allowing more resources to be allocated to foraging, mating, and development [85,86]. Such effects are known to be particularly pronounced in winter for insects, when thermal stress is typically higher, and the relative benefits of small temperature increases can be substantial [87].

The detritivores, arthropods that feed on decaying matter, did not show significant differences between the treatments. The rate of detrital decomposition depends on the quality of the detritus and the physical environment, including factors such as temperature and water availability [56]. In the Azores, precipitation is highest from November to January, and it varies considerably within each month, depending on elevation and longitude. Humidity is generally high, with median relative humidity (RH) values around 80%, and the cooler months are typically January and February [88], with temperatures above 0 °C in our experiment. Therefore, detritivores may be particularly dependent on the availability of decaying matter and less affected by temperature changes than other guilds.

For the beetles, increased temperatures impacted most of the functional traits studied, with stronger effects also observed during winter. This suggests that beetles may be more sensitive to temperature changes during colder months, potentially due to metabolic and physiological constraints. Ectotherms, such as beetles, rely on external sources of heat to regulate their body temperature, and their metabolic rates are strongly influenced by ambient temperatures [85,86]. At lower temperatures, metabolic rates are typically slower, potentially limiting activity levels and overall fitness [89]. Therefore, even small increases in temperature during winter could have a disproportionately large impact on beetle physiology and behavior, aligning with findings from other studies demonstrating the vulnerability of ectotherms to temperature changes, particularly during colder periods [87].

During the winter season, the detritivore arthropod abundance may have been lower within the OTCs compared to the natural control plots due to several factors. Temperature variation within OTCs can disrupt the stable microhabitat that detritivores rely on [90]. While OTCs are designed to elevate overall temperatures, the enclosed environment may lead to greater daily and seasonal fluctuations in temperature compared to the surrounding natural plots [91]. This could have adversely affected the physiology and behavior of these moisture-dependent organisms, making the environment less suitable for their survival and reproduction. In addition to temperature, OTCs may also alter moisture levels within leaf litter. By trapping heat, OTCs can potentially dry out the organic substrate, reducing the humidity levels that detritivores require [92]. Alternatively, the OTCs may have created a more condensed environment, leading to excessive moisture, which is also detrimental to detritivore communities. Furthermore, the physical presence of the OTC structures themselves could have interfered with the natural accumulation and distribution of organic matter, which serves as the primary food source for detritivore arthropods. This disruption of the microhabitat could have limited the availability of resources and led to a decreased abundance of these decomposer organisms within the OTCs compared to the control plots.

#### 4.2.2. Ecological Traits

Interestingly, while the beetle body size peaked at middle elevations in both summer and winter, a larger average body size was observed within the OTCs during the summer. This contradicts the general trend of smaller body sizes in ectotherms under warmer temperatures, as suggested by [42,93]. This discrepancy could be attributed to species-specific responses to temperature, altered resource availability within the OTCs, or interactions with other environmental factors not explored in this study. For instance, increased temperatures might have led to a greater quality and amount of food for certain beetle species [94]. 

The reduced activity of the nocturnal beetles, as well as the reduced activity of the diurnal and crepuscular arthropods and beetles within the OTC treatments during the winter, may indicate a means of avoiding thermal stress, as suggested by Thiele [70]. However, this result needs to be considered with caution, as in our case, the temperature within the OTCs during the winter was warmer than in the control plots, but not extreme. Therefore, it should be considered whether other factors, such as food availability, could have been responsible for the observed effect, and not only thermal stress, which should be low at this time of year. Pawar et al. (2024) [95] also highlighted that thermal adaptation in arthropods is constrained by the temperature of peak performance for key life-history traits, which could also explain our results.

More low-dispersal arthropods were found in the OTCs than in the control plots. The opposite was found for the high-dispersal arthropods, which were more abundant in the control plots than in the OTCs. Thus, the high dispersers may have left the OTCs when the abiotic conditions became unsuitable, with only low dispersers remaining. This pattern is consistent with the finding that as climate change occurs, high dispersers are the first to seek out better abiotic conditions [96,97]. 

During the summer, larger beetles were more prevalent in the OTCs, potentially indicating that increased temperatures within OTCs favor the presence of some larger beetle species. This finding opposes the prevailing trend in ectotherms, which tend to have smaller body sizes at warmer temperatures [42], suggesting species-specific responses to temperature or altered resource dynamics within OTCs.

#### 4.2.3. Feeding Behavior

The study found that the feeding behavior of arthropods, including piercing, sucking, chewing, cutting, and external digestion, was not impacted by increased temperatures within the OTCs. Although mouthparts are unique to each arthropod kingsofamily, this finding is also consistent with several studies suggesting that while temperature can affect metabolic rates and activity levels, the abundance of different arthropod feeding strategies often remains constant across temperature variations [98,99,100].

#### 4.2.4. Beetle Color and Iridescence

The data showed an increased number of iridescent beetles inside the OTCs at lower elevations during winter, suggesting that elevated temperatures enhance the prevalence of iridescence. The increased presence of iridescent beetles inside the OTCs implies that iridescence may provide a thermoregulatory benefit under higher temperatures, aiding in heat reflection and reducing thermal stress [76,79]. This ability to reflect sunlight and maintain lower body temperatures would be particularly advantageous during warmer periods, supporting the survival and activity of iridescent beetles [101]. 

The higher abundance of black beetles in the OTCs during winter suggests potential thermal benefits associated with darker coloration. Darker colors can absorb more heat, providing significant thermal advantages in colder environments [80,81]. This adaptation probably enhanced the survival and activity of the black beetles during the winter months, when ambient temperatures are lower. The observation of fewer black beetles in the control plots compared to the OTCs supports the hypothesis that the increased temperatures within the OTCs would create more favorable conditions for these beetles in winter. Such findings align with the theory that melanism confers thermal benefits by facilitating heat absorption, which is crucial for ectotherms to maintain metabolic processes in cooler climates [83]. The reddish-brown coloration might offer a balance between camouflage and thermal regulation, providing sufficient heat absorption without the risk of overheating in moderate environments [82,102,103].

These findings contribute to our understanding of how beetles adapt to varying microclimatic conditions and offer a basis for further research on the ecological functions of beetle coloration and iridescence.

## 5. Conclusions

Overall, this study highlights the complex interplay between elevation, temperature, and season in shaping arthropod and beetle communities. The observed patterns emphasize the need to consider multiple environmental factors when assessing the impact of climate change on biodiversity. The functional traits approach unveiled more patterns among the studied arthropod communities in response to climatic changes than the taxonomical approach previously applied by Wallon et al. (2023) [30]. Other studies comparing the taxonomical and the functional approach showed the importance of considering functional traits in analyzing arthropod communities and agreed that the functional approach enhances knowledge of the processes underlying patterns of diversity in complex ecological communities involving multiple trophic levels [104,105,106]. However, analyses including functional traits appear to be more complex to interpret due to various factors and interactions [107]. 

Future research should further explore the underlying mechanisms driving these patterns and their implications for ecosystem functioning and resilience. The results of this study contribute to this growing body of knowledge and highlight the importance of examining arthropod responses at the community level and considering functional traits, rather than focusing solely on individual species.

In conclusion, the observed effects of increasing temperature and elevational gradients on the functional traits of arthropods and beetles highlight the complex and functional-group-specific nature of thermal responses within these communities.

## Figures and Tables

**Figure 1 insects-15-00677-f001:**
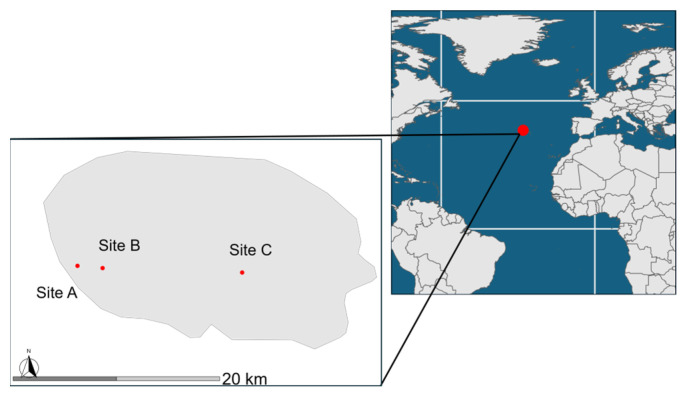
Localization of the island of Terceira and experimental sites.

**Figure 2 insects-15-00677-f002:**
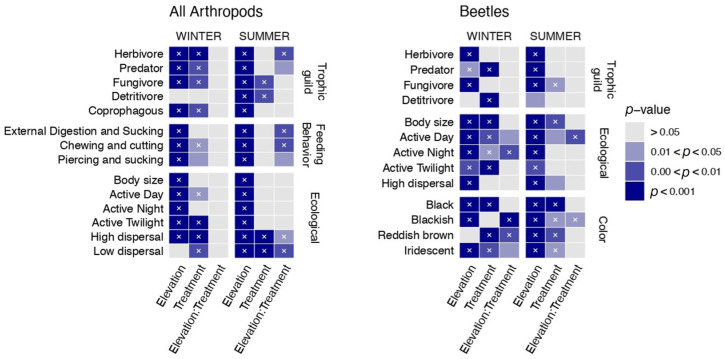
Heatmap for winter and summer season, highlighting the results of the analysis of variance (ANOVA), considering all arthropods and only beetles, which was conducted to investigate the effects of elevation, treatment, and their interaction (Elevation:Treatment) on each trait community weighted means (CWM) variables. On the (**left**) *y*-axis are the different traits analyzed. IngEDS: external digestion and sucking; IngCC: chewing and cutting; IngPS: piercing and sucking; Hdisp: high dispersal ability; Ldisp: low dispersal ability; ActDay: diurnal; ActNig: nocturnal; ActTw: active during twilight. On the (**right**) *y*-axis are the different categories of traits analyzed. The *x*-axis presents the different factors analyzed: elevation, treatment, and their interaction (Elevation:Treatment). The colors indicate the level of significance (dark blue, *p* < 0.001; medium blue, 0.00 < *p* < 0.01; light blue, 0.01 < *p* <0.05, and gray, *p* > 0.05). Cells marked with a “×” mean that the effect remained significant after the FDR correction for multiple testing. Results of the ANOVA are detailed in Table A1 and Table A3.

**Figure 3 insects-15-00677-f003:**
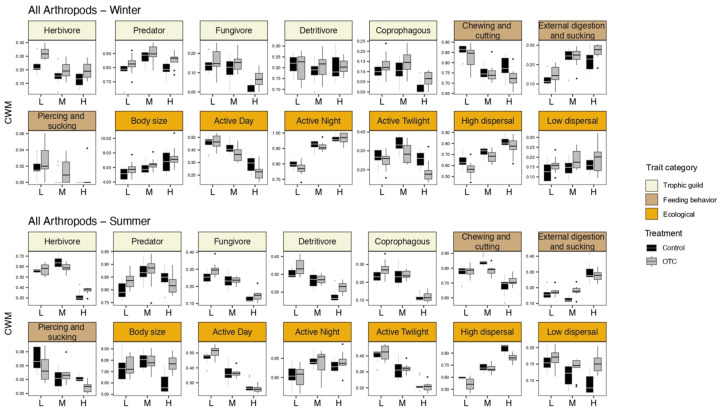
Boxplots highlighting the effect of the treatment (control vs. OTCs) on different traits along the elevation gradient considering all arthropods during the winter and the summer. On *y*-axis are the values of the community weighted means (CWM) and on *x*-axis is the elevation factor: low (L), middle (M), and high (H) elevation. Black box plots indicate the control plots, while gray box plots represent the OTCs. ANOVA and post hoc pairwise comparisons results are available in Table A1 and Table A2.

**Figure 4 insects-15-00677-f004:**
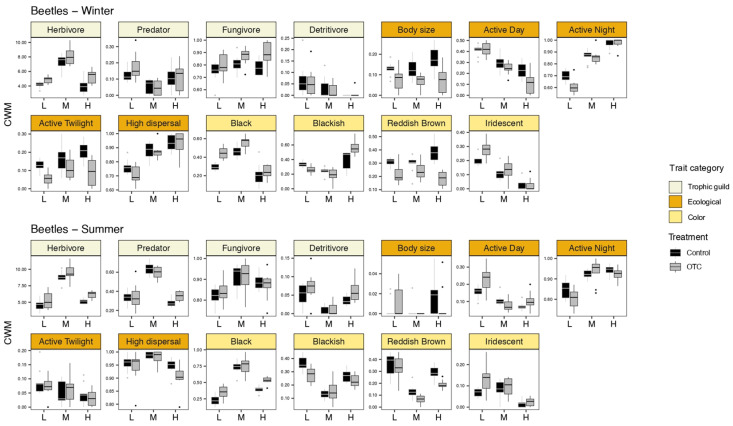
Boxplots highlighting the effect of the treatment (control vs. OTCs) on different traits along the elevation gradient considering only beetles during the winter and the summer. On *y*-axis are the value of the community weighted means (CWM) and on *x*-axis is the elevation factor: low (L), middle (M), and high (H) elevation. Black box plots indicate the control plots, while gray box plots represent the OTCs. ANOVA and post hoc pairwise comparisons results are available in Table A3 and Table A4.

**Table 1 insects-15-00677-t001:** Selection, definition, and ecological relevance of the functional traits used in the study.

Traits Category	Traits	Data Type	Attributes (Abbreviations)	Definition	Ecological Relevance
Guild	Type of food	Multi-choice nominal	Plants (FoodPl); animals (FoodAni); fungi (FoodFg); detritus (FoodDet); coprophagous organisms (FoodCopro)	Each guild is categorized according to the primary food consumed by the species during their adult stages, except for Lepidoptera, for which the classification is based on the feeding habits of the larvae.	Under different climatic conditions, the natural resources available for arthropod feeding may vary. This can affect arthropod feeding guilds and lead to different arthropod communities.
Feeding behavior	Mode of ingestion	Nominal	Chewing and cutting (IngCC); piercing and sucking (IngPS); external digestion and sucking (IngEDS)	Depends on the type of mouthpart and, thus, on the way in which arthropods feed. For Lepidoptera, the classification is based on the mouthparts of the larvae.	Determines feeding strategies and diet specialization, influencing nutrient cycling and energy flow within ecosystems.
Ecological	Body size	Continuous	Standardized body size (bodysizeStand); body size (bodysize)	Defined as species mean body length.	Species’ body size is linked to their metabolism, which influences their adaptability and ecological niches.
Ecological	Dispersal ability	Nominal	Low dispersal ability (Ldisp); High dispersal ability (Hdisp)	Dispersal ability (high or low) is attributed based on physical characteristics of the animal. Presence or absence of wings and ballooning capacity for spiders.	Animals with high dispersal ability will tend to respond more quickly to unsuitable abiotic conditions in order to find a more suitable environment. In contrast, animals with low dispersal ability will have more difficulty in migrating to better conditions.
Ecological	Diel activity	Multi-choice nominal	Day (ActDay); Night (ActNig); Twilight (ActTw)	Refers to the main period of activity of the species during the day.	Diel activity shapes how species interact with their environment and each other, and also impacts how arthropods respond to heat. Nocturnal species are less affected by daytime temperatures, as they avoid direct sunlight and extreme heat by remaining hidden.
Coloration	Color of beetle cuticle	Nominal	Black (Color_Black); blackish (Color_Blackish); reddish-brown (Color_ReddishBrown)	Refers to the main nuance observed on the cuticle. Black refers to a black cuticle. Blackish refers to a main black tendency with a reddish/orange color. Reddish-brown refers to a main tendency orange/brown.	Color can serve several functions, including signaling, mating, camouflage, and thermoregulation. For this reason, beetle color should provide information about the heat tolerance of the species.
Coloration	Presence of iridescence on beetle cuticle	Nominal	Iridescence (IridescentColor)	Refers to the presence or absence of iridescence on the beetle cuticle.	The presence of iridescence helps thermoregulation by reducing solar absorption and limiting risks of overheating.

## Data Availability

Data used for this publication are available at: Wallon, S.; Elias, R.B.; Borges, P.A.V. (2023). Monitoring grassland arthropods in an in situ climate change experiment (Terceira, Azores, Portugal). Version 1.5. Universidade dos Açores. Sampling event dataset, https://doi.org/10.15468/u2xh5g accessed on 19 July 2024, via GBIF.org.

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
