# Peer review of "Unveiling Arthropod Responses to Climate Change: A Functional Trait Analysis in Intensive Pastures"

_insects, 2024, doi:10.3390/insects15090677_

Round 1

Reviewer 1 Report

Comments and Suggestions for Authors

The study investigates the impact of elevated temperatures on arthropod communities using Open Top Chambers (OTCs) to simulate increased temperatures, along with the functional traits of ground-dwelling arthropods. The research was well-planned, and a substantial amount of data was obtained.

The data were analyzed appropriately.

I have only a few minor points that should be addressed before publication:

1. **Definition of "detritivores":** The author discusses detritus derived only from plant materials. If animal detritus is also relevant, it should be included in the discussion. If detritivores that feed on animal detritus are not addressed, a definition should be provided in the earlier part of the manuscript. The results section should mention if no animal-detritus feeders were captured, while the Methods and Materials section should indicate if these feeders were excluded from the detritivore analysis.

2. **Lines 299-311:** This section duplicates information found in the Methods and Materials part. I suggest deleting this redundancy.

3. **Lines 353-359:** This content should be moved to the subsection "3.1.2. Ecological Traits."

4. **Line 367:** The responses appear to differ between the interior and exterior of the OTCs.

5. **Line 370:** There seem to be differences in the treatments at the highest altitude during the summer.

6. **Figure 4:** The term "Predatore" should be corrected to "Predator."

7. **Lines 527-528:** The statement “while those utilizing external digestion increased” is not adequately supported by the results mentioned in the main Results section, except for Appendix A.

8. **Lines 550-556:** The statement regarding blackish beetles being dominant at high elevation (L550-551) and their highest abundances at low elevations due to more effective thermoregulation (L554-556) appears contradictory. These points do not coincide with each other.

9. **Lines 601-602:** The phrase “the increase in temperature inside the OTC may be beneficial for their metabolism and create a favorable environment” seems inconsistent compared to the preceding part of the same paragraph and the previous paragraph. Additional explanation is needed for clarity.

10. **Line 622:** Please clarify what "Higher" refers to—does it indicate higher values compared to those outside the OTC or in relation to high-dispersing arthropods?

11. ** Figures 2, 3, and 4** It is better to be adjusted by grouping related categories for each function. This approach will improve clarity and facilitate readers' understanding of the information presented.

Author Response

Thank you for taking the time to provide such valuable feedback on our manuscript. Your insightful comments have greatly contributed to the improvements made in this version. We have carefully reviewed and addressed each of your suggestions (please see our responses in blue). All revisions are highlighted in yellow in the manuscript.

The study investigates the impact of elevated temperatures on arthropod communities using Open Top Chambers (OTCs) to simulate increased temperatures, along with the functional traits of ground-dwelling arthropods. The research was well-planned, and a substantial amount of data was obtained.

The data were analyzed appropriately.

Thank you very much for your comments.

I have only a few minor points that should be addressed before publication:

Comments 1: 1. **Definition of "detritivores":** The author discusses detritus derived only from plant materials. If animal detritus is also relevant, it should be included in the discussion. If detritivores that feed on animal detritus are not addressed, a definition should be provided in the earlier part of the manuscript. The results section should mention if no animal-detritus feeders were captured, while the Methods and Materials section should indicate if these feeders were excluded from the detritivore analysis.

Response 1: Thank you for highlighting this aspect which was not clear.

In our experiment, all the decomposers were considered. The discussion focused more on organic matter of plant origin, especially in summer. In the species we collected, most were not strictly decomposers of organic matter of either animal or plant origin, but of both.

So even if we have considered all the detritivores, it is true that during the summer a drop in their abundance is mainly due to the quasi-absence of litter, which represents a considerable part of the diet of our detritivores. We have updated the discussion to make it clear that we considered all detritivores and not just those that feed on organic matter of plant origin. See lines 451-458.

We have also added references:

Haitao, W., Xianguo, L., Ming, J., & Xiao, B. (2009). Impacts of soil fauna on litter decomposition at different succession stages of wetland in sanjiang plain, China. Chinese Geographical Science, 19(3), 258–264. https://doi.org/10.1007/s11769-009-0258-y

Mancinelli, G., Sangiorgio, F., & Scalzo, A. (2013). The effects of decapod crustacean macroconsumers on leaf detritus processing and colonization by invertebrates in stream habitats: A meta-analysis. International Review of Hydrobiology, 98(4), 206–216. https://doi.org/10.1002/iroh.201301539

Comments 2: 2. **Lines 299-311:** This section duplicates information found in the Methods and Materials part. I suggest deleting this redundancy.

Response 2: Thank you for your comment. It has been considered and this section has been deleted. Very minor changes were made in the Materials and Methods section to clarify the separation of different types of traits (e.g., trophic level, feeding behavior, and ecological traits). See now the lines 241-243 and lines 254-255.

Comments 3: 3. **Lines 353-359:** This content should be moved to the subsection "3.1.2. Ecological Traits."

Response 3: Yes, you are right. Your advice has been taken into consideration. See now the lines 349-355

Comments 4: 4. **Line 367:** The responses appear to differ between the interior and exterior of the OTCs.

Response 4: Indeed, responses seems to differ between the interior and exterior of the OTCs. But the ANOVA performed to investigate the effects of the elevation and the treatment on functional traits of all arthropods including the interaction terms (Elev:Treatment) on each CWM variables did not reveal significant differences (Table A1).

Season

Trait

Factor

Df

Sum Sq

Mean Sq

F value

Pr(>F)

p.adjust

Summer

Body size

Elevation

2

2350.5

1175.3

61.207

<0.001

<0.001

Treatment

1

61.2

61.2

3.187

0.08

0.116

Elev:Treatment

2

102

51

2.657

0.079

0.116

Residuals

54

1036.9

19.2

Winter

Body size

Elevation

2

809.3

404.6

21.583

<0.001

<0.001

Treatment

1

15

15

0.801

0.375

0.477

Elev:Treatment

2

66.7

33.3

1.778

0.179

0.309

Residuals

54

1012.4

18.7

We slightly modify the text to be more accurate. See lines 359-361

Comments 5: 5. **Line 370:** There seem to be differences in the treatments at the highest altitude during the summer.

Response 5: As mentioned in the previous point (4), even if it seems to have a difference between treatment at high altitude during the summer, ANOVA performed did not revealed significant differences.

Comments 6: 6. **Figure 4:** The term "Predatore" should be corrected to "Predator."

Response 6: Thank you for pointing out this error. It has been corrected.

Comments 7: 7. **Lines 527-528:** The statement “while those utilizing external digestion increased” is not adequately supported by the results mentioned in the main Results section, except for Appendix A.

Response 7: References to the Appendix A (Table A1 and Table A2) were added to support this statement in the Results section. See lines 372.

Comments 8: 8. **Lines 550-556:** The statement regarding blackish beetles being dominant at high elevation (L550-551) and their highest abundances at low elevations due to more effective thermoregulation (L554-556) appears contradictory. These points do not coincide with each other.

Response 8: As it was written, this part was indeed contradictory. However, the statement that blackish beetles were dominant at high altitudes referred to endemic Azorean beetles, living at higher elevation than our experiment. No endemic beetles were collected during the study. The statement was made mainly to emphasize the fact that endemic beetles living in the higher part of the island do not show iridescence. We have rewritten this part of the text in order to be more explicit in the message we want to convey and to avoid any contradiction. See lines 549 -557

Comments 9: 9. **Lines 601-602:** The phrase “the increase in temperature inside the OTC may be beneficial for their metabolism and create a favorable environment” seems inconsistent compared to the preceding part of the same paragraph and the previous paragraph. Additional explanation is needed for clarity.

Response 9: Thank you for pointing out this flaw in the manuscript. Our results indicate the opposite, and the discussion led to a misinterpretation. We have updated this part to be accurate and to explain our results. See lines 603-620.

We have also added references:

Bardgett, R. D., Freeman, C., & Ostle, N. J. (2008). Microbial contributions to climate change through carbon cycle feedbacks. ISME Journal, 2(8), 805–814. https://doi.org/10.1038/ismej.2008.58

Bordier, C., Dechatre, H., Suchail, S., Peruzzi, M., Soubeyrand, S., Pioz, M., Pélissier, M., Crauser, Di., Conte, Y. Le, & Alaux, C. (2017). Colony adaptive response to simulated heat waves and consequences at the individual level in honeybees (Apis mellifera). Scientific Reports, 7(1). https://doi.org/10.1038/s41598-017-03944-x

Lutterschmidt, W. I., & Hutchison, V. H. (1997). The critical thermal maximum: Data to support the onset of spasms as the definitive end point. Canadian Journal of Zoology, 75(10), 1553–1560. https://doi.org/10.1139/z97-782

Thanks a lot for your input!

Comments 10: 10. **Line 622:** Please clarify what "Higher" refers to—does it indicate higher values compared to those outside the OTC or in relation to high-dispersing arthropods?

Response 10: "Higher" refers to the fact that there are more low dispersers in the OTCs than in the control plots. The sentence has been rewritten to avoid confusion. See lines 641-643

Comments 11: 11. ** Figures 2, 3, and 4** It is better to be adjusted by grouping related categories for each function. This approach will improve clarity and facilitate readers' understanding of the information presented.

Response 11: Thanks for your advice. On the Figure 2, we add the categories on the left y-axis and add this information in the legend. For Figures 3 and 4, we now add different color coding to separate the different functional groups and make the figures easier to understand.

Reviewer 2 Report

Comments and Suggestions for Authors

This study was quite broad in scope.  The time and effort invested was likely significant, which may be the reason for only analyzing one gradient transect.  Replication at each elevation was good, but it was still only one study site.  Would it be possible to repeat the study at other locations within the Azores? 

Some isolation of the elevation variable was incorporated by focusing on intensively managed fields.  Was there variation within these habitats?

The authors mentioned that the endemic arthropod fauna was greatly diminished due to land management, and did not distinguish between endemic and exotic species within the functional groups.  Is it possible that any of the loss of endemics could be due to climate change?

Functional traits of adult species – it is often the immature stage that does the majority of the feeding, so this might be a consideration if focusing on different feeding strategies.

The authors acknowledge the challenges that come with interpreting effects on a functional grouping.  Functional groupings often underscore the inherent variation within that grouping.  As long as that is clearly stated, it is a good approach.

Comments on the Quality of English Language

Minor grammatical issues:

64:  “several” species of arthropods is probably not accurate

67: stronger “events”

77:  at the same time “open”

114:  “erbivore” should be “herbivores”

337: The x axis “presents”

545:  End of the sentence seems to be missing?

683:  “…underline the complex and species specific nature of thermal responses in these communities.”  This article does not really describe impacts on species, but rather functional groupings.

Author Response

We appreciate the time and effort that you have dedicated to provide insightful comments and feedback on the manuscript, they directly led to this improved current version of the manuscript. We have carefully considered the questions and comments and tried our best to address each of them (see responses in blue). All changes are highlighted in yellow in the manuscript.

Comments 1: This study was quite broad in scope.  The time and effort invested was likely significant, which may be the reason for only analyzing one gradient transect.  Replication at each elevation was good, but it was still only one study site.  Would it be possible to repeat the study at other locations within the Azores? 

Response 1: Thank you for your valuable comments. Regarding the experiment, we had three pastures with 20 plots each, of which 10 were control plots and 10 were OTC plots, giving a total of 10 replications per treatment nested within each altitude, which we considered sufficient for the analysis. This experimental design was considered with an appropriate statistical analysis (see current Lines 279-289). Although it would be nice to duplicate the experiment in other locations within the Azores, some difficulties have to be taken into account:

- First, this kind of experiments are logistically complex and it is difficult to convince farmers to give up their pastures for the time of the experiment (plots and OTCs remained in the same location throughout the year for both sampling) and thus not to have cattle grazing. This is a loss of income for them. We were very lucky to have the collaboration of Virginia Pires, a student at the University of the Azores, who convinced her father to borrow two of their pastures (the third pasture belongs to the Experimental Farm of the University of the Azores.).

- Second, since the fieldwork was quite extensive (setting 240 traps and then collecting them), we would need a larger team or more collaborators in other islands to be able to include other transects. We hope that this will be possible in the future in order to strengthen our current findings.

Comments 2: Some isolation of the elevation variable was incorporated by focusing on intensively managed fields.  Was there variation within these habitats?

Response 2: During the experiment, each field was managed in the same way. The only difference was the plant cover. Although all three fields were predominantly grass, the low and medium elevation fields were covered with Italian ryegrass, Lolium multiflorum Lam. (Poaceae) and the high-altitude pasture was covered with common velvet grass Holcus lanatus L. (Poaceae). This is due to the way intensive pastures are managed here in the Azores. Italian ryegrass is more adapted to lower altitudes, while velvet grass is more adapted to higher altitudes. Based on the elevation of their field, farmers seed the most suitable grasses. However, despite the difference in the grass species sown (see Lines 139-140), all the three sites were managed in the same way (set up of the pitfall traps at the same period and absence of livestock). For contrast, the semi-natural pastures are diverse in endemic and native plant species and the arthropod community structure is different (see Cardoso et al., 2009).

Cardoso, P., Lobo, J.M., Aranda, S.C., Dinis, F., Gaspar, C. & Borges, P.A.V. (2009). A spatial scale assessment of habitat effects on arthropod communities of an oceanic island. Acta Oecologica-International Journal of Ecology, 35: 590-597

Comments 3: The authors mentioned that the endemic arthropod fauna was greatly diminished due to land management, and did not distinguish between endemic and exotic species within the functional groups.  Is it possible that any of the loss of endemics could be due to climate change?

Response 3: Lines 196-198, in the Introduction section, we mentioned the fact that the presence of endemic species is reduced due to significant intensification regime and land disturbance and that we decided not to include the biogeographical origin of species. It is a fact that contrarily to the semi-natural pastures not considered in our study, the Azorean intensive pastures, with few exceptions, are deprived of endemic species. For example, in our experiment, among the 171 morphospecies collected, only one was endemic and 29 native non-endemics to the Azores. Exotic arthropods were indeed dominant in intensive pastures (see Cardoso et al. 2009 above). Lhoumeau & Borges (2023) highlighted the fact that over a period of 10 years of sampling, endemic arthropods remained stable in their habitat (native forest) and are mostly associated and dependent on the native vegetation, which is their ecological niche. Thus, the loss of endemic species in intensive pasture is due to land disturbance (absence of native vegetation to rely on) and intensive management, regardless of climatic conditions.

Lhoumeau, S.; Borges, P.A.V. Assessing the Impact of Insect Decline in Islands: Exploring the Diversity and Community Patterns of Indigenous and Non-Indigenous Arthropods in the Azores Native Forest over 10 Years. Diversity 2023, 15, 753. https://doi.org/10.3390/d15060753

Comments 4: Functional traits of adult species – it is often the immature stage that does the majority of the feeding, so this might be a consideration if focusing on different feeding strategies.

Response 4: We agree with you that in many arthropod groups it is often the immature stages that do the majority of the feeding, but this can vary depending on the specific group and the ecological role they play. For example, our study focused on ground-dwelling arthropods and consisted primarily of beetles, arachnids, centipedes, millipedes, hemipterans.

For beetles, feeding patterns vary greatly. In some species, particularly herbivorous or wood-boring beetles like weevils and bark beetles, larvae are the primary feeders. However, in predatory beetles—which were predominant in our study—both larvae and adults are active feeders.

In arachnids, it's typically the adult stage that is most active in feeding. Both immature and adult arachnids are predators, but adults tend to have a broader diet and are more effective hunters.

Centipedes are predatory throughout their life cycle, but adult centipedes, being larger, are usually more active hunters and consume more prey.

Millipedes, as detritivores, feed consistently from immature stages through adulthood, consuming large quantities of decaying organic matter.

Hemipterans, represented by 13 species in our study, also feed during both immature and adult stages, often on plant sap or other fluids, with substantial feeding occurring in both stages.

A special attention was given to Lepidoptera in which larvae are clearly the primary feeders. Although we only had three species with only 20 occurrences over the three fields and both seasons, we define the guild and feeding behavior based on the larval stage and not the adult.  They were also not considered as a focus group in our study.

We have already mentioned in Table. 1 for the guilds that "Each guild is categorized according to the primary food consumed by the species during their adult stages, except for Lepidoptera where the classification is based on the feeding habits of the larvae".  We have now updated the text in Table. 1 for the feeding behavior to make it clear how we have defined it: "Depends on the type of mouthpart and thus on the way the arthropods feed. For Lepidoptera, the classification is based on the mouthparts of the larvae".

Given that we were collecting mostly adult specimens at certain times of the year (e.g., winter and summer), and given the species composition of our samples, we agreed that considering adult stages was relevant to our study.

Comments 5: The authors acknowledge the challenges that come with interpreting effects on a functional grouping.  Functional groupings often underscore the inherent variation within that grouping.  As long as that is clearly stated, it is a good approach.

Response 5: Thank you for your observation.

Comments on the Quality of English Language

Minor grammatical issues:

Comments 6: 64:  “several” species of arthropods is probably not accurate

Response 6: We change the word “several” by “various”. There are a few hundred species, although we cannot give an explicit number. This is why we speak of "various arthropod species". See line 64.

Comments 7: 67: stronger “events”

Response 7: Thank you for spotting this error. We have corrected it. See line 67.

Comments 8: 77:  at the same time “open”

Response 8: Thank you for reporting this error. We have now fixed it. See line 77

Comments 9: 114:  “erbivore” should be “herbivores”

Response 9: Thank you for pointing out this mistake. We’ve made the necessary correction. See line 114.

Comments 10: 337: The x axis “presents”

Response 10: Thank you for reporting this grammatical error. We have corrected it.

Comments 11: 545:  End of the sentence seems to be missing?

Response 11: Thank you for notifying this typo error. We've made the necessary correction. See line 544.

Comments 12: 683:  “…underline the complex and species specific nature of thermal responses in these communities.”  This article does not really describe impacts on species, but rather functional groupings.

Response 12: Following your recommendation, we have updated the text to be more accurate. See lines 704-706